Role of necroptosis in pulpitis: integration of bacterial infection, immune imbalance, and oxidative stress

Wang Xuefei 1
Wang Yaying 2
Pang Zhenyu 2
Gong Peiyao 2
Ma Xuyi 2
Qiu Yuhao 2
Li Jianzhen 2
Peng Lianjie penglianjie@zju.edu.cn lianjiepeng@163.com 3
Liu Zhichao zhichaoliu@zju.edu.cn 393628269@qq.com 3
1 Stomatology Hospital, School of Stomatology, Zhejiang University School of Medicine , Hangzhou , Zhejiang Province , China
2 Changsha Medical University , Changsha , Hunan Province , China
3 Stomatology Hospital, School of Stomatology, Zhejiang University School of Medicine, Zhejiang Provincial Clinical Research Center for Oral Diseases, Key Laboratory of Oral Biomedical Research of Zhejiang Province, Cancer Center of Zhejiang University , Hangzhou , Zhejiang Province , China
Haraguchi Tokuko
Electronic publication date: 2025 Oct 20
Publication date: 2025
Volume: 13
Electronic Location ID: e20209
Received 2025 Mar 11; Accepted 2025 Sep 18
Copyright: ©2025 Wang et al.
Copyright year: 2025
Copyright holder: Wang et al.
License: This is an open access article distributed under the terms of the Creative Commons Attribution License, which permits unrestricted use, distribution, reproduction and adaptation in any medium and for any purpose provided that it is properly attributed. For attribution, the original author(s), title, publication source (PeerJ) and either DOI or URL of the article must be cited.
License URL: https://creativecommons.org/licenses/by/4.0/

Keywords: Necroptosis, Pulpitis, Bacterial infection, Immune imbalance, Oxidative stress

Funding: The authors received no funding for this work.

==============================
Background

Pulpitis is a common oral inflammatory condition driven by bacterial infection, immune imbalance, and oxidative stress, often involving pro-inflammatory cell death within the dental pulp. Necroptosis—a regulated, caspase-independent form of cell death mediated by the receptor-interacting protein kinase 1 (RIPK1), RIPK3, and mixed-lineage kinase domain-like protein (MLKL)—has garnered growing interest in various infectious and inflammatory diseases. However, its specific role in pulpitis remains underexplored.

Aim of the study

This review aims to explore how bacterial infection, immune imbalance, and oxidative stress synergistically activate necroptosis, and proposes for the first time that excessive activation of necroptosis may contribute to the progression of pulpitis.

Methodology

A narrative review was conducted using PubMed, Web of Science, and Google Scholar. Searches employed the keywords “pulpitis”, “necroptosis”, and related MeSH terms, combined with Boolean operators.

Results

Based on a comprehensive review of the existing literature, this review is the first to propose that the integration of bacterial infection, immune imbalance, and oxidative stress may contribute to the progression of pulpitis from reversible inflammation to irreversible necrosis.

Conclusion

Bacterial infection in pulpitis may activate necroptosis through the Toll-Like Receptor 4 (TLR4)-RIPK3 pathway, leading to the release of damage-associated molecular patterns (DAMPs) that disrupt immune homeostasis, while mitochondrial dysfunction-induced ROS further aggravates oxidative stress. These interacting mechanisms may collectively exacerbate pulpal inflammation and tissue damage, ultimately resulting in irreversible pulpitis. Accordingly, targeting necroptosis pathways may offer a promising therapeutic approach for pulpitis.

Introduction

Necroptosis is a pro-inflammatory form of programmed cell death that has gained increasing attention in the fields of infectious and inflammatory diseases. Its core signaling pathway involves the receptor-interacting protein kinase 1 (RIPK1), RIPK3, and mixed-lineage kinase domain-like protein (MLKL) complex (Bao, Ye & Ren, 2022). Upon activation of death receptors such as tumor necrosis factor receptor (TNFR), Toll-like receptor 3/4 (TLR3/4), and Fas cell surface death receptor (Fas), protein kinase 3 (RIPK3) phosphorylates MLKL, which triggers its translocation to the plasma membrane, where it induces membrane rupture and the release of intracellular pro-inflammatory mediators (Bedoui, Herold & Strasser, 2020). This form of cell death not only contributes directly to tissue damage but also promotes immune cell recruitment via damage-associated molecular patterns (DAMPs), perpetuating a vicious cycle of inflammation-driven cell death (Bock & Tait, 2020).

Pulpitis, a common oral inflammatory condition, is characterized by persistent inflammation, intense pain, and progressive tissue damage (Byers & Närhi, 1999). In the conventional model of pulpitis, bacterial invasion activates classical inflammatory pathways such as the TLR4/myeloid differentiation primary response 88 (MyD88) cascade, leading macrophages and fibroblasts to release pro-inflammatory mediators that, in turn, induce oxidative stress (Zhang, Cui & Li, 2021). However, necroptosis may serve as a critical link between these pathological processes. Bacterial infection not only activates the TLR4-RIPK3 pathway via pathogen-associated molecular patterns (PAMPs) but also disrupts local immune homeostasis, promotes macrophage polarization toward the proinflammatory M1 phenotype, elevates tumor necrosis factor-alpha (TNF-α) production, and triggers Tumor Necrosis Factor Receptor 1 (TNFR1)-dependent necroptosis signaling (Weinlich et al., 2017; Brenner, Blaser & Mak, 2015). The release of DAMPs caused by immune imbalance aggravates TLR/nuclear factor kappa B (NF-κB) signaling through positive feedback, driving more cells into necroptosis (Pullerits et al., 2003). Mitochondrial dysfunction may result in reactive oxygen species (ROS) accumulation, potentially aggravating oxidative stress and promoting inflammatory damage in dental pulp tissue (Adameova et al., 2022).

The specific role of necroptosis in pulpitis remains insufficiently explored. Based on a comprehensive review of the existing literature, this review aims to explore how bacterial infection, immune imbalance, and oxidative stress synergistically activate necroptosis, and thereby proposes for the first time that excessive activation of necroptosis may contribute to the progression of pulpitis from reversible inflammation to irreversible necrosis. Accordingly, targeting necroptosis pathways may offer a promising therapeutic approach for pulpitis.

Survey Methodology

Databases and search platforms

PubMed, Web of Science, Scopus, and Google Scholar were systematically searched to ensure broad coverage of high-quality scientific literature across biomedical and interdisciplinary domains. To ensure reliability, only articles published in established academic journals were included from Google Scholar.

Search strategy and keywords

A structured search strategy was employed, integrating Medical Subject Headings (MeSH) and free-text keywords to capture both standardized and emerging terminology. Key terms included “pulpitis,” “dental pulp inflammation,” “necroptosis,” “RIPK1,” “RIPK3,” “MLKL,” “oxidative stress,” “bacterial infection,” and “immune response.” Boolean operators (AND, OR) were applied to enhance both the sensitivity and specificity of the search, facilitating the identification of core and related studies on necroptosis in pulpitis.

Inclusion and exclusion criteria

Eligible sources included original research articles, systematic reviews, and meta-analyses published in academic journals. We excluded studies focusing solely on apoptosis or pyroptosis, or if they did not address the roles of bacterial infection, immune imbalance, and oxidative stress in necroptosis and its involvement in pulpitis. Editorials, opinion pieces, and reports lacking experimental or clinical evidence were also omitted to ensure scientific rigor. The included studies primarily investigated how the synergistic interplay of these factors contribute to necroptosis in pulpitis and potentially drive the progression from reversible inflammation to irreversible necrosis.

Strategies to minimize bias

To minimize potential bias in this narrative review, multiple databases were searched to reduce source bias, and studies presenting divergent findings were included to reflect a range of perspectives. Citation bias was addressed by critically assessing the relevance and methodological quality of eligible studies. Particular attention was given to the design, sample size, and reproducibility of the included studies to ensure the reliability of the review findings.

Mechanisms of Necroptosis

Necroptosis and apoptosis

Programmed cell death (PCD) is a tightly regulated biological process orchestrated by both intracellular and extracellular signals, and encompasses distinct subtypes such as apoptosis and necroptosis (Chen et al., 2023). Apoptosis is characterized by maintained membrane integrity and the absence of an inflammatory response, primarily mediated by caspase-8 activation. In contrast, necroptosis is typically triggered when caspase-8 is inhibited, leading to plasma membrane rupture and a potent pro-inflammatory response through the RIPK1-RIPK3-MLKL signaling axis (Murphy & Vince, 2015).

A complex cross-regulatory interplay exists between apoptosis and necroptosis. Caspase-8 activation cleaves RIPK1 and RIPK3, thereby suppressing necroptosis. Conversely, when caspase-8 is inactive, RIPK1 interacts with RIPK3 to form necrosomes, driving necroptotic cell death (Tummers & Green, 2017). RIPK1 functions as a critical molecular switch in determining cell death fate; in apoptosis, it promotes non-inflammatory cell clearance via the Fas-Associated protein with Death Domain (FADD)-caspase-8 pathway, while in necroptosis, altered ubiquitination enhances its interaction with RIPK3, initiating pro-inflammatory signaling cascades (Feltham & Silke, 2017). Mitochondria play distinct roles in these pathways; during apoptosis, they release cytochrome c to initiate the caspase cascade, whereas in necroptosis, mitochondrial ROS activate RIPK3 and establish a positive feedback loop that amplifies necroptotic signaling (Deragon et al., 2020). The plasticity of cell death mechanisms holds significant implications for inflammatory diseases. In Salmonella Typhimurium-induced enteritis, caspase-8 activation suppresses necroptosis in intestinal epithelial cells, thereby preserving barrier integrity and restricting pathogen invasion (Hefele et al., 2018) (Table 1).

Table 1 Comparison of necroptosis and apoptosis.

Category	Apoptosis	Necroptosis	
Key molecules	Caspase 8/9	RIPK1, RIPK3, MLKL	
Membrane integrity	Maintained	Disrupted	
Inflammatory response	None	Present (release of DAMPs)	
Role of mitochondria	Release of cytochrome c → activation of caspases	Release of ROS → activation of RIPK3	
Effect of Caspase 8	Initiation of apoptosis and inhibition of necrosis	Necroptosis was induced in absence of caspase 8 activity	
Significance of pathology	Remove the cells	Inflammatory tissue damage	

The extrinsic pathway of necroptosis

TNF/TNFR1 pathway

The extrinsic pathway of necroptosis is primarily triggered by interactions between immune cells and damaged or infected target cells, with TNFR1 acting as the prototypical death receptor (Gill, Tsung & Billiar, 2010). Following TNF-α binding to TNFR1, TNF receptor-associated death domain (TRADD) recruits RIPK1, initiating the assembly of membrane-bound complex I (Zhang et al., 2018). Regulated by the deubiquitinase cylindromatosis (CYLD), RIPK1 dissociates from the membrane and suppresses sustained NF-κB signaling. It subsequently contributes to the assembly of cytosolic complex II, where it undergoes autophosphorylation (Yuan, Amin & Ofengeim, 2019). This phosphorylation enables the recruitment and activation of RIPK3 via RIP homotypic interaction motif (RHIM)-mediated binding. Activated RIPK3 subsequently phosphorylates MLKL at its pseudokinase domain (PsKD), promoting its oligomerization and translocation to the plasma membrane. At the plasma membrane, phosphorylated MLKL compromises membrane integrity, resulting in cell swelling, membrane rupture, and the release of DAMPs, thereby promoting necroptosis and amplifying inflammatory responses in various pathological contexts (Green, 2019).

Fas and TLR pathways

In addition to TNFR1, other death receptors also contribute to necroptosis regulation. For example, Fas ligand (FasL) binds to its receptor Fas to promote necrosome formation and induce necroptosis in the absence of caspase-8. Similarly, TLRs, particularly TLR3 and TLR4, recognize PAMPs and DAMPs and activate the adaptor protein TIR-domain-containing adapter-inducing interferon-β (TRIF), which interacts with RIPK3 via RHIM-mediated binding to promote necroptotic signaling under caspase-8-deficient conditions (Hahn & Liewehr, 2007).

The intrinsic pathway of necroptosis

Intrinsic triggers and necrosome assembly

Intrinsic necroptosis is regulated by cellular responses to microenvironmental stress and can be triggered by oxidative stress, metabolic disturbances, or genotoxic insults (Lee et al., 2013; Murphy, 2009; Van den Berghe et al., 2014; Ingram et al., 2019). Oxidative stress disrupts the mitochondrial electron transport chain, leading to excessive ROS accumulation that promotes RIPK3 activation via a positive feedback loop. Adenosine triphosphate (ATP) depletion impairs (cIAP1/2) activity, inhibiting RIPK1-dependent survival signaling and facilitating necroptosis. Genotoxic stress, such as DNA double-strand breaks, activates Z-DNA Binding Protein 1 (ZBP1), which directly binds and activates RIPK3 independent of RIPK1. Under these stressors, RIPK3 phosphorylation is a central event that drives necrosome assembly by interacting with RIPK1 or ZBP1, leading to MLKL phosphorylation and execution of intrinsic necroptosis (Yang et al., 2018).

Mitochondrial dysfunction

Mitochondrial dysfunction is tightly regulated by modifications of outer mitochondrial membrane proteins (MOMPs) (Hasan et al., 2023; Hoeger et al., 2023). Under oxidative stress or mitochondrial dysfunction, gasdermin D N-terminal domain (GSDMD-N) inserts into the outer mitochondrial membrane, forming pores that release mitochondrial reactive oxygen species (mtROS) and lead to cellular swelling, membrane rupture, and amplified inflammation. Meanwhile, receptor-interacting RIPK3 activates phosphoglycerate mutase family member 5 (PGAM5) isoforms, with PGAM5 short isoform dephosphorylating dynamin-related protein 1 (Drp1) at Ser637 to enhance its GTPase activity, inducing mitochondrial fission and further ROS release (Bang, Miki & Kang, 2023).

ROS amplify necroptotic signaling through a calcium/calmodulin-dependent protein kinase II (CaMKII)/mitochondrial permeability transition pore (mPTP)-driven feedback loop (Zhang et al., 2016). ROS activate CaMKII via oxidation and Thr286 autophosphorylation, sustaining its activity. CaMKII then opens the mPTP, causing mitochondrial swelling, cristae disruption, and further ROS release (Li & Shen, 2022). Concurrently, CaMKII enhances RIPK3 activity, promoting MLKL oligomerization, membrane translocation, and rupture. The resulting DAMPs release activates TLR4/NF-κB signaling and upregulates TNF-α, which further stimulates ROS production via TNFR1, perpetuating necroptosis and inflammation (Chen et al., 2023; Choi et al., 2019; Zhang et al., 2017) (Fig. 1).

Figure 1 Molecular mechanisms of necroptosis.

TNF-α, tumor necrosis factor alpha; FAS, fas receptor; FASL, fas ligand; TNFR1, tumor necrosis factor receptor 1; TLRs, toll-like receptors; ROS, reactive oxygen species; TRADD, TNFR1-associated death domain protein; TRAF2/5, TNF receptor-associated factors 2/5; CIAP1/2, cellular inhibitors of apoptosis protein 1/2; mPTP, mitochondrial permeability transition pore; CYLD, cylindromatosis; RIPK1, receptor-interacting protein kinase 1; RIPK3, receptor-interacting protein kinase 3; MLKL, mixed lineage kinase domain-like protein; GSDMD, gasdermin D; GSDMD-N, gasdermin D N-terminal domain; CaMKII, calcium/calmodulin-dependent protein kinase II; Caspase-8, cysteine aspartate-specific protease 8; TRADD, TNFR1-associated death domain protein; PGAM5, phosphoglycerate mutase 5; TRIF, TIR-domain-containing adapter-inducing interferon- β. Upon TNF-α stimulation, TNFR1 activates a signaling cascade that may lead to necroptosis, depending on the composition of downstream protein complexes. Binding of TNF-α to TNFR1 promotes the assembly of Complex I, which is comprised of TRADD, RIPK1, TNF Receptor-Associated Factor 2/5 (TRAF2/5), and cIAP1/2. When RIPK1 is deubiquitinated at lysine 63 by CYLD, Complex I transitions into Complex IIa, which includes RIPK1, RIPK3, FADD, caspase-8, and TRADD. Alternatively, in the absence of deubiquitination, Complex I forms Complex IIb—TRADD-independent and composed of RIPK1, RIPK3, FADD, and caspase-8. Complexes IIa and IIb represent critical decision nodes for cell fate, determining whether apoptosis or necroptosis ensues. Caspase-8 serves as a key regulator in this process; its activation cleaves RIPK1 and RIPK3, steering the cell toward apoptosis. Conversely, caspase-8 inhibition or deficiency facilitates the formation of the necrosome, consisting of phosphorylated RIPK1, RIPK3, and MLKL. Phosphorylated MLKL translocates to the plasma membrane, disrupts membrane integrity, and induces cell swelling, lysis, and the release of intracellular contents—hallmarks of necroptosis. Mitochondria play a central role in amplifying necroptotic signaling. Phosphorylated PGAM5 destabilizes mitochondrial outer membrane proteins, increasing membrane permeability and promoting excessive ROS production. Additionally, phosphorylated CaMKII contributes to necroptosis by activating ion channels, which increases intracellular concentrations of Ca2+, Na+, and H+, leading to osmotic imbalance and membrane rupture. It also promotes the opening of the mPTP, resulting in ROS leakage into the cytoplasm. The resulting ROS burst further activates CaMKII, reinforcing a positive feedback loop that amplifies necroptotic signaling. Moreover, GSDMD, a member of the gasdermin family, exacerbates mitochondrial dysfunction during necroptosis. Upon cleavage by caspase-1 under pro-inflammatory conditions, GSDMD forms 10–14 nm pores in the mitochondrial outer membrane, causing substantial ROS release into the cytoplasm and further amplifying the necroptotic cascade.

Therefore, necroptosis can be triggered via extrinsic or intrinsic mechanisms, which differ in initiating factors and regulatory pathways. The extrinsic pathway is primarily mediated by death receptors such as TNFR1, Fas, and TLRs, in response to pathogens or inflammatory signals. In contrast, the intrinsic pathway is activated by intracellular stressors, including oxidative stress, metabolic imbalance, and genotoxic damage, often involving mitochondrial dysfunction and excessive ROS generation. Although both pathways converge on RIPK3-mediated MLKL activation and membrane disruption, their upstream signaling events are distinct. This review primarily focuses on the extrinsic pathway in the pathogenesis of pulpitis, while also considering the potential contribution of oxidative stress through intrinsic mechanisms.

Inflammation in Necroptosis

Physiological functions of necroptosis in anti-infective immunity and tissue homeostasis

Necroptosis plays a vital role in innate immune defense by eliminating intracellular pathogens, particularly when apoptosis is suppressed, such as during viral inhibition of caspase activity (Mocarski, 2023; Yu et al., 2024). In such cases, necroptosis restricts pathogen replication by sacrificing infected cells and activating immune responses. Impairment of this pathway can compromise host defense, facilitating pathogen spread. In Staphylococcus aureus infection models, necroptosis is essential for bacterial control; wild-type mice suppress bacterial dissemination via necroptotic cell death, whereas inhibition of RIPK1 or deletion of MLKL increases bacterial loads and inflammation (Kitur et al., 2016). Beyond infection, necroptosis also contributes to epithelial renewal by clearing damaged or senescent cells, thereby maintaining barrier integrity in tissues like the intestinal epithelium (Günther et al., 2013).

While necroptosis plays physiological roles in anti-infective immunity and tissue homeostasis, its aberrant activation may contribute to inflammatory responses and tissue injury. This review primarily discussed its pro-inflammatory effects in the progression of pulpitis.

Pro-inflammatory role of necroptosis in immune responses via the extrinsic pathway

Bacterial infection-induced necroptosis

Necroptosis, a pro-inflammatory form of PCD marked by plasma membrane rupture and release of intracellular contents, plays a key role in bacterial pathogenesis. In refractory apical periodontitis, Enterococcus faecalis induces necroptosis in macrophages through the RIPK3-MLKL axis. Pharmacological inhibition of RIPK3 significantly reduces E. faecalis-induced periapical inflammation and alveolar bone loss, highlighting RIPK3 as a critical driver of disease progression (Dai et al., 2022).

PAMPs, conserved microbial motifs, are recognized by host pattern recognition receptors (PRRs), particularly TLRs. In the absence or inhibition of caspase-8, TLR3 and TLR4 can mediate PAMP-induced necroptosis (Kaczmarek, Vandenabeele & Krysko, 2013). Lipopolysaccharide (LPS), a key PAMP from Gram-negative bacteria, is a potent necroptosis inducer implicated in infectious disease progression. In sepsis, Escherichia coli-derived LPS activates RIPK1, RIPK3, and MLKL in intestinal epithelial cells, causing severe barrier disruption—an effect attenuated by RIPK1 inhibition using necrostatin-1 (Nec-1) (Liu et al., 2021). Additionally, LPS also induces necroptosis in macrophages via TLR4 signaling (Legarda et al., 2016).

TLRs, key components of the innate immune system, recognize both PAMPs and DAMPs. Upon sensing ligands such as LPS or double-stranded RNA (dsRNA), TLRs recruit the adaptor protein TRIF, which interacts with RIPK3 to promote MLKL phosphorylation. Activated MLKL disrupts the plasma membrane, leading to DAMP release. These DAMPs further activate TLRs on neighboring cells, amplifying the inflammatory response. Additionally, TLR4 activates the NF-κB pathway via MyD88, increasing TNF-α production and inducing necroptosis through the TNFR1-RIPK1-RIPK3 cascade (Fan et al., 2016).

Amplification of inflammation by necroptosis

Necroptosis plays a central role in initiating, sustaining, and exacerbating inflammation by facilitating the release of pro-inflammatory mediators and DAMPs. As a form of regulated cell death, necroptosis results in plasma membrane rupture and the subsequent extracellular release of intracellular contents, including high mobility group box 1 (HMGB1), mitochondrial DNA (mtDNA), and pro-inflammatory cytokines such as interleukin-1β (IL-1β) and TNF-(Zhang et al., 2018; Lacević, Vranić & Zulić, 2004). These molecules are detected by PRRs on neighboring cells, activating downstream pathways such as NF-κB and amplifying the inflammatory response (Roh & Sohn, 2018). In conditions like irreversible pulpitis, IL-1β and TNF-α are markedly upregulated in necrotic pulp tissue, contributing to local inflammatory activation (Kokkas et al., 2007). TNF-α, a pleiotropic cytokine, binds to TNFR1 and activates the RIPK1-RIPK3-MLKL axis, directly inducing necroptosis and establishing a feed-forward “necrosis-inflammation-necrosis” loop. In inflammatory bowel disease (IBD), M1-polarized macrophages primarily contribute to mucosal barrier disruption and increased pathogen invasion through TNF-α secretion. This, in turn, reinforces M1 polarization, forming a pro-inflammatory positive feedback loop that exacerbates chronic intestinal inflammation (Lissner et al., 2015).

Beyond cytokine signaling, DAMPs released during necroptosis recruit neutrophils and macrophages through chemokines such as C-X-C Motif Chemokine Ligand 8 (CXCL8) and C-C Motif Chemokine Ligand 2 (CCL2) (Sprooten et al., 2020). The resulting immune cell infiltration sustains cytokine production and increases the risk of secondary necrosis, further propagating DAMP release and driving a self-perpetuating cycle of chronic inflammation. Thus, the sustained and dysregulated release of DAMPs during necroptosis emerges as a critical mechanism underlying the progression of chronic inflammatory diseases (Fig. 2).

Figure 2 Necroptosis-driven inflammation in pulpitis progression.

DAMPs, damage-associated molecular patterns; IL-α, interleukin-alpha; IL-1β, interleukin-1 beta; IL-2, interleukin-2. During necroptosis, membrane disruption, organelle swelling, and nuclear fragmentation lead to the release of DAMPs, which activate PRRs on neighboring cells and trigger local inflammation. This recruits immune cells such as neutrophils and macrophages, which release cytokines like Interleukin-2 (IL-2), IL-1β, and TNF-α, further amplifying necroptotic signaling. As the cycle persists, widespread immune and tissue cell death may aggravate pulp inflammation, impair reparative capacity, and promote infection spread to the periapical region, ultimately resulting in irreversible pulpitis.

Dual identity of ROS: the pro-inflammatory mechanism in intrinsic necroptosis

ROS play a dual role in necroptosis, acting as both upstream triggers and downstream effectors, and are critically involved in initiating and amplifying inflammatory responses (Liu et al., 2023a). Under necroptosis-induced mitochondrial dysfunction, excessive ROS accumulation activates signaling cascades such as NF-κB and the NOD-like receptor family pyrin domain-containing 3 (NLRP3) inflammasome, leading to the upregulation of pro-inflammatory cytokines, including TNF-α and IL-6, and contributing to a localized inflammatory microenvironment (Pang et al., 2021). In rheumatoid arthritis, for instance, TNF-α and mitochondrial ROS synergistically induce synovial cell necroptosis, exacerbating joint inflammation and tissue destruction (Meng, Wei & Shan, 2024).

ROS not only serve as intrinsic signaling molecules but also intersect with extrinsic necroptotic pathways. Mitochondrial ROS generated downstream of TNF/TNFR1 signaling enhance RIPK1 activation, while the RIPK1-RIPK3 complex reciprocally amplifies mitochondrial ROS production, forming a positive feedback loop via the “TNFR1-mitochondria-RIPK3” axis (Lossi, 2022). In addition, the importance of ROS in inducing necroptosis has also been verified in an in vivo model of tuberculosis-infected zebrafish (Roca & Ramakrishnan, 2013).

Studies have demonstrated that mitochondria-targeted antioxidants, such as MitoVit E, significantly inhibit NF-κB activation, underscoring the pivotal role of ROS in necroptosis-associated inflammation (Hughes, Murphy & Ledgerwood, 2005). Additionally, in ischemia-reperfusion injury-induced acute kidney injury (IRI-AKI), mitochondrial ROS not only aggravate necrosis but also impair mtDNA integrity by disrupting mitochondrial Mitochondrial Transcription Factor A (TFAM) activity, sustaining chronic inflammation (Zhao et al., 2021). Collectively, ROS emerge as central regulators linking inflammation and necroptosis, driving pathogenesis in inflammatory and degenerative diseases.

Necroptosis in pulpitis

Pulpitis is a common oral inflammatory condition primarily driven by bacterial infection. Microbial invasion of the pulp triggers a strong immune response that can ultimately lead to pulp necrosis (Moroishi et al., 2015). In the following sections, this review will, for the first time, discuss how bacterial infection, immune imbalance, and oxidative stress synergistically activate necroptosis, which may contribute to the progression of pulpitis from reversible inflammation to irreversible necrosis.

Mechanisms of bacteria-driven necroptosis in pulpitis

Oral inflammatory diseases, including pulpitis, are primarily caused by Gram-positive and Gram-negative bacteria that establish a pro-inflammatory microenvironment conducive to biofilm formation (Neve, Corrado & Cantatore, 2014). During cariogenic progression, acid metabolites produced by specific bacterial groups demineralize enamel, allowing Gram-positive bacteria to penetrate further, initiate cavity formation, and degrade dentin. Streptococcus, Lactobacillus, and Actinomyces dominate the oral microbiota and can invade pulp tissue through dentinal tubules, with deeper lesions accelerating infection (Pasparakis & Vandenabeele, 2015). Studies have shown that Streptococcus infection activates RIPK3-dependent necroptosis via MLKL, leading to mitochondrial ROS accumulation, mPTP opening, and exacerbated inflammation (Huang et al., 2021). Interestingly, Gram-positive bacteria such as MRSA can induce necroptosis in alveolar macrophages by releasing extracellular vesicles enriched with TNF-α and miR-146a-5p, which activate RIPK1, RIPK3, and MLKL phosphorylation (Bai et al., 2024). In contrast, Gram-negative infections trigger necroptosis through the activation of TLR4, TNFR, and type I IFN receptor signaling in macrophages (Robinson et al., 2012). These findings suggest that bacterial infection may contribute to the pathogenesis of pulpitis by activating necroptosis.

PAMPs, conserved microbial motifs, are likely to initiate immune responses by engaging PRRs on host cells. Research has shown that PAMPs, such as LPS from Gram-negative bacterial cell walls, activate the TLR4/MyD88/NF-κB signaling pathway, thereby upregulating NLRP3 inflammasome components and pro-IL-1β expression in human dental pulp fibroblasts (HDPFs) (Zhang et al., 2015). As reversible pulpitis progresses to irreversible pulpitis, enhanced NLRP3 activation promotes inflammatory cytokine production and exacerbates tissue inflammation. In parallel, LPS also induces IL-8 gene transcription and protein release in human dental pulp stem cells via the same TLR4-MyD88-NF-κB axis, further amplifying the inflammatory response (He et al., 2013). Both immune and non-immune cells in dental pulp express PRRs, including TLRs and NOD-like receptors (NLRs), which recognize PAMPs and initiate innate immune responses (Shi et al., 2015). Furthermore, human odontoblast-like cells contribute to inflammation by activating TLR2 in response to bacterial infection, leading to the release of pro-inflammatory cytokines such as IL-8 and IL-6 (Farges et al., 2011). Therefore, bacteria-induced necroptosis may contribute to the initiation and progression of pulpitis.

Regulation of necroptosis in pulpitis immune imbalance

The innate immune system is the primary defense against bacterial invasion and tissue damage, functioning through host cell recognition of pathogenic agents (Song et al., 2023). However, the confined anatomy of dental pulp limits phagocytic clearance before bacterial exposure. Inflammation is typically initiated when oral microbiota penetrate the pulp via caries, trauma, dentin cracks, apical foramina, or dentinal tubules, leading to vascular alterations, immune cell infiltration, and progressive tissue damage (Simpson et al., 2022). Gingival tissue is comprised of a variety of immune and non-immune cells, including macrophages, neutrophils, and odontoblasts. Odontoblasts, as the first line of defense against cariogenic bacteria, play a key role in regulating pulpal immune and inflammatory responses to dentin-penetrating pathogens (Zhou et al., 2021). Macrophages, derived from circulating monocytes, exhibit functional diversity shaped by the local microenvironment (Tancharoen et al., 2014). In the early stages of pulpitis, macrophages predominantly display a pro-inflammatory M1 phenotype, which gradually shifts toward an anti-inflammatory M2 phenotype as the disease progresses. This transition coincides with a shift from innate to adaptive immunity, characterized by T cell-mediated secretion of IL-2, IL-4, and IL-13 (Simpson et al., 2022). Additionally, B cells may contribute to the adaptive response in pulpitis, as indicated by elevated levels of IgG1, IgA, and IgE in carious pulpal tissue, suggesting active humoral involvement.

Necroptosis may contribute significantly to the pathogenesis of pulpitis by promoting the release of cytokines and DAMPs, which in turn activate immune cells and amplify inflammatory responses. Among various cytokines, TNF-α plays a central role in regulating pulp cell function and mediating inflammation by upregulating the expression of tissue-type plasminogen activator (tPA) and matrix metalloproteinases (MMPs) in human dental pulp cells and gingival fibroblasts, thereby facilitating pulp tissue degradation (Ueda & Matsushima, 2001). Furthermore, autocrine TNF-α release promotes vasodilation, leukocyte recruitment, and additional pro-inflammatory cytokine secretion, thereby intensifying inflammation and accelerating disease progression (Tang et al., 2023). DAMPs, released by stressed or damaged pulp cells, bind to TLRs and elicit potent immune responses. In pulpitis patients, HMGB1 levels are markedly elevated compared to healthy pulp tissue. HMGB1 interacts with the receptor for advanced glycation end products (RAGE) and enhances TLR2/4 signaling, while also promoting CpG DNA delivery and TLR9 activation, further driving cytokine production (Mahmoudi et al., 2017).

In pulpitis, necroptosis-induced DAMPs may disrupt immune homeostasis and trigger cytokine release, which in turn reinforces necroptosis, forming a self-perpetuating inflammatory loop that drives the progression from reversible to irreversible pulpitis.

Oxidative stress-driven necroptosis in pulpitis

Pulpitis induced by dental caries is closely associated with oxidative stress, mitochondrial dysfunction, and cell death (Vaseenon et al., 2023). Current clinical interventions often fail to promote effective pulp repair, with oxidative stress posing a major therapeutic challenge (Vengerfeldt et al., 2017). Mitochondrial impairment under oxidative stress represents a key pathological mechanism in pulpitis. Although oxidative stress-induced inflammation initially aims to eliminate pathogens and support tissue repair, disrupted mitochondrial dynamics may impair pulp homeostasis and exacerbate tissue injury (Wang et al., 2025).

Necroptosis and its associated ROS production appear to be critical contributors to pulpal inflammation. ROS act as pro-inflammatory mediators by activating signaling pathways such as p38 MAPK and NF-κB, thereby stimulating excessive cytokine release from immune and resident pulp cells, which intensifies inflammation and tissue damage (Wang et al., 2023). Experimental studies have demonstrated that hydrogen peroxide (H2O2)-based bleaching agents generate ROS in dental pulp stem cells, resulting in cytotoxicity and nociceptive signaling (Chen et al., 2021). Similarly, 2-hydroxyethyl methacrylate (HEMA), a monomer used in restorative materials, can diffuse through dentinal tubules to the pulp, where it induces ROS production, eliciting inflammatory and cytotoxic responses (Orimoto, Kitamura & Ono, 2022). ROS accumulation further exacerbates pulpal damage by promoting inflammatory mediator release and prostaglandin synthesis, increasing pain sensitivity.

Therefore, necroptosis may exacerbate oxidative stress and tissue injury in irreversible pulpitis by disrupting mitochondrial function, thereby reinforcing a self-perpetuating cycle of inflammation and cell death that drives disease progression.

Synergistic interplay of bacterial infection, immune imbalance and oxidative stress

In the pathological progression of pulpitis, bacterial infection, immune imbalance, and oxidative stress may act synergistically to amplify the pro-inflammatory effects of necroptosis. In the early stages of infection, bacterial invasion through the dentin barrier leads to the release of microbial metabolites such as LPS, which activates necroptotic signaling via TLR4. This initiates innate immune responses, recruits immune cells (e.g., macrophages and neutrophils), and promotes the secretion of pro-inflammatory cytokines such as TNF-α and IL-1β. LPS stimulation drives macrophage polarization toward a pro-inflammatory M1 phenotype, which enhances TNF-α release and ROS production to combat pathogens (Tan et al., 2016). However, excessive ROS disrupt mitochondrial function and promote necroptosis by oxidatively modifying RIPK3 and CaMKII, thereby enhancing RIPK1-RIPK3-MLKL necrosome assembly and activation (Chen et al., 2023).

Simultaneously, DAMPs released from necroptotic cells—such as HMGB1 and mitochondrial DNA—stimulate surrounding immune cells via TLR9 or the receptor for RAGE, leading to chemokine release (e.g., IL-6, CXCL8) and further neutrophil recruitment (Mahmoudi et al., 2017). The influx of immune cells amplifies ROS production, establishing a self-perpetuating ROS-immune activation-necroptosis feedback loop (Ziehr & MacDonald, 2024). Mitochondrial ROS additionally activate the NLRP3 inflammasome, inducing macrophage pyroptosis and further IL-1β secretion, thereby exacerbating inflammation. This synergy between pyroptosis and necroptosis accelerates irreversible pulp tissue damage (Casey et al., 2025). In the hypoxic environment of deep carious lesions, impaired mitochondrial respiration shifts energy metabolism toward glycolysis, resulting in lactate accumulation and acidification. These conditions compromise antioxidant defenses and may drive the progression from reversible inflammation to irreversible pulp necrosis (Solaini et al., 2010) (Fig. 3 and Table 2).

Figure 3 Role of necroptosis in pulpitis: integration of bacterial infection, immune imbalance and oxidative stress.

dsRNA, double-stranded RNA; LPS, lipopolysaccharide; PAMPs, pathogen-associated molecular patterns; TLRs, toll-like receptors; PRR, pattern recognition receptors; MyD88, myeloid differentiation primary response protein 88; TRIF, TIR-domain-containing adapter-inducing interferon-β; NF-κ B, nuclear factor kappa B; TNF-α, tumor necrosis factor alpha; TNFR1, tumor necrosis factor receptor 1; DAMPs, damage-associated molecular patterns; IL-2, interleukin-2; IL-4, interleukin-4; RIPK1, receptor-interacting protein kinase 1; RIPK3, receptor-interacting protein kinase 3; MLKL, mixed lineage kinase domain-like protein; GSDMD, gasdermin D; GSDMD-N, gasdermin D N-terminal domain; GSDMD-C, gasdermin D C-terminal domain; Caspase-1, cysteine-dependent aspartate-directed protease 1; CaMKII, calcium/calmodulin-dependent protein kinase II; mPTP, mitochondrial permeability transition pore; Drp1, dynamin-related protein 1; PGAM5, phosphoglycerate mutase 5; DSPP, dentin sialophosphoprotein; MAPK, mitogen-activated protein kinase; ROS, reactive oxygen species. Bacterial infection may initiate necroptosis in pulpitis via the TLR4–RIPK3 pathway triggered by PAMPs such as LPS and dsRNA. These PAMPs are recognized by TLR4, which activates TRIF-dependent signaling to form the TRIF–RIPK3 complex, leading to RIPK3 phosphorylation, MLKL activation, and necroptotic cell death. Membrane rupture releases DAMPs that stimulate TLRs on neighboring cells, amplifying inflammation. In parallel, TLR4 also activates the NF-κ B pathway via MyD88, enhancing TNF-α production and further promoting necroptosis through TNFR1 signaling. DAMP release from necroptotic cells may disrupt immune homeostasis and contribute to dysregulated inflammation in pulp tissue. Odontoblasts exposed to cariogenic bacteria exhibit impaired reparative function, including reduced dentin matrix synthesis, alkaline phosphatase (ALP) activity, and DSPP expression. Immune responses in pulpitis are regulated by both innate and adaptive cells. Early stages are dominated by M1-polarized macrophages, which later transition to M2 phenotypes. As disease progresses, T and B cells contribute by secreting pro-inflammatory cytokines (e.g., IL-2, IL-4) and immunoglobulins (e.g., IgG1, IgA), respectively. Necroptosis exacerbates inflammation by releasing DAMPs and cytokines such as TNF-α, IL-1β, and IL-6, potentially creating a self-amplifying loop that drives the shift from reversible to irreversible pulpitis. Necroptosis also induces mitochondrial dysfunction and oxidative stress. GSDMD forms pores in mitochondrial membranes, causing ROS leakage and membrane rupture. RIPK3 activates CaMKII and PGAM5, triggering mPTP opening and mitochondrial fragmentation, which further increases ROS accumulation and disrupts cellular homeostasis, potentially promoting pulp cell death. Collectively, the interaction among bacterial infection, immune imbalance, and oxidative stress may form a pathogenic network that facilitates the transition of pulpitis from a reversible to an irreversible stage.

Necroptosis: a novel therapeutic target in pulpitis

Pulpitis, one of the most prevalent oral diseases, typically results from the exposure of dental pulp to external stimuli, leading to severe pain and eventual pulp necrosis if left untreated. As inflammation progresses, it can extend beyond the pulp, leading to apical periodontitis (Weindel et al., 2022). In recent years, vital pulp therapy (VPT) has emerged as a promising alternative to root canal treatment (RCT) by aiming to preserve pulp vitality. However, the clinical application of VPT remains challenging, particularly in caries-induced pulpitis, due to the difficulty of accurately evaluating the extent of inflammation and effectively controlling it to facilitate tissue healing (Moroishi et al., 2015). Given that necroptosis may play a pivotal role in the pathogenesis of pulpitis, we propose for the first time that targeting necroptosis pathways may offer a promising therapeutic approach for its treatment.

Table 2 Summary of included studies on necroptosis in pulpitis.

Number	Study content	Research model	References	
1	Bacterial invasion activates TLR4/MyD88 and other pathways through PAMPs to induce the release of pro-inflammatory factors, which in turn triggers oxidative stress and mitochondrial dysfunction	Traditional pulpitis model: Human dental pulp cells (hDPCs) were stimulated with LPS to simulate the inflammatory conditions of pulpitis	Zhang, Cui & Li (2021)	
2	Imbalance of immune regulation leads to the release of DAMPs, which aggravates TLR/NF-κB signaling through positive feedback and drives more cells into programmed necrosis	RA model: An HMGB-1–driven RA animal model was established by intra-articular injection of 1–5 µg recombinant HMGB-1 into the knee joints of mice	Pullerits et al. (2003)	
3	Enterococcus faecalis-induced macrophage necroptosis through the RIPK3-MLKL pathway, and inhibition of RIPK3 reduces bone loss	Mouse model of refractory apical periodontitis: Refractory apical periodontitis was established by infecting the apical region of the mandibular first molar in mice with E. faecalis, which induced macrophage necroptosis via the RIPK3/MLKL pathway	Dai et al. (2022)	
4	In sepsis, Escherichia coli-derived LPS activates RIPK1, RIPK3, and MLKL in intestinal epithelial cells, causing severe barrier disruption—an effect attenuated by RIPK1 inhibition using necrostatin-1 (Nec-1)	Sepsis-associated intestinal injury model: Intestinal injury was induced in piglets by systemic administration of LPS	Liu et al. (2021)	
5	Macrophages undergo necroptosis in response to LPS stimulation through TLR4 signaling pathway	In vitro necroptosis model using mouse bone marrow-derived macrophages (BMDMs): Necroptosis was induced by stimulating BMDMs with LPS under conditions of caspase-8 inhibition, which promoted autocrine TNF production and activated the RIPK3/MLKL pathway	Legarda et al. (2016)	
6	TLR4 activates the NF-κB pathway through a myd88-dependent mechanism and induces an increase in TNF-α, which induces necroptosis via TNFR1	Mouse spinal cord injury (SCI) model: Mechanical injury was applied to the spinal cord in mice to induce RIPK3/MLKL-mediated necroptosis of astrocytes via activation of the TLR4/MyD88 signaling pathway	Fan et al. (2016)	
7	In IBD, M1 macrophages secrete TNF-α to impair the mucosal barrier and promote pathogen invasion, thereby reinforcing M1 polarization and exacerbating chronic inflammation	Co-culture model of human monocytes/M1 macrophages and intestinal epithelial cells: Human peripheral blood CD14+ monocytes were polarized in vitro into M1 or M2 macrophages and subsequently co-cultured with intestinal epithelial cell lines	Lissner et al. (2015)	
8	Mitochondrial ROS form a positive feedback loop through the CaMKII/mPTP pathway to amplify necroptosis	Murine models of myocardial necroptosis induced by ischemia-reperfusion and doxorubicin: mice are subjected to cardiac ischemia-reperfusion or doxorubicin injection to induce oxidative stress–related cardiomyocyte necrosis and to evaluate the role of the RIP3–CaMKII pathway in myocardial injury	Zhang et al. (2016)	
9	Abnormal accumulation of ROS can activate the NF-κB and NLRP3 signaling axes, up-regulate proinflammatory factors, and create a local inflammatory microenvironment	Low-dose CdTe quantum dot (QD)-induced hepatotoxicity model: mice are intraperitoneally injected with low-dose CdTe QDs to induce ROS production and simulate oxidative stress–mediated liver injury caused by nanomaterial exposure	Pang et al. (2021)	
10	TNF-α and mitochondrial ROS synergistically promote necrotizing apoptosis of synovial cells and aggravate joint destruction	Fibroblast-like synoviocyte (FLS) model: FLSs isolated from patients with rheumatoid arthritis (RA) are cultured and stimulated with TNF-α	Meng, Wei & Shan (2024)	
11	Mitochondria-targeted antioxidants, such as MitoVit E, significantly inhibit NF-κB activatio, underscoring the pivotal role of ROS in necroptosis-associated inflammation	Mitochondrial ROS-regulated TNF-α-induced apoptosis model in U937 cells: U937 cells are treated with TNF-α in combination with the mitochondria-targeted antioxidant MitoVit E to investigate the temporal regulation of NF-κB activation by mitochondrial ROS	Hughes, Murphy & Ledgerwood (2005)	
12	In ischemia-reperfusion injury, mitochondrial ROS aggravates cell necrosis, destroys mtDNA stability, and initiates persistent inflammation	Ischemic acute kidney injury (AKI) mouse model: mice are subjected to renal ischemia-reperfusion to induce kidney injury	Zhao et al. (2021)	
13	Streptococcal infection activates MLKL via RIPK3, triggering ROS accumulation and triggering necroptosis	Streptococcus pneumoniae infection model in mice:Streptococcus pneumoniae is administered intratracheally into the lungs of mice.	Huang et al. (2021)	
14	Gram-positive bacterial infection exacerbates pneumonia through necroptosis, releasing exosomes of TNF-α and miR-146a-5p	EV-MRSA-induced in vitro necroptosis model: murine alveolar macrophage cell line (MH-S) is infected with MRSA, followed by isolation and purification of extracellular vesicles (EVs) released from the infected cells	Bai et al. (2024)	
15	Gram-negative bacterial infection induces necroptosis through activation of TLR4, TNFR, etc	IFN-I–macrophage necroptosis model: bone marrow–derived macrophages (BMDMs) are infected with Salmonella Typhimurium and treated with type I interferon-neutralizing antibodies	Robinson et al. (2012)	
16	PAMPs, such as LPS from Gram-negative bacterial cell walls, activate the TLR4/MyD88/NF-κB signaling pathway, thereby upregulating NLRP3 inflammasome components and pro-IL-1β expression in human dental pulp fibroblasts (HDPFs)	Human dental pulp fibroblast model: human dental pulp fibroblasts are cultured in vitro and stimulated with LPS and APT to mimic an inflammatory environment associated with bacterial infection and cellular injury	Zhang et al. (2015)	
17	LPS also induces IL-8 gene transcription and protein release in human dental pulp stem cells via the same TLR4–MyD88–NF-κB axis, further amplifying the inflammatory response	Human dental pulp stem cell (hDPSC) model: dental pulp tissue is isolated from impacted human third molars and cultured in vitro to obtain hDPSCs, which are then stimulated with lipopolysaccharide (LPS) to induce an inflammatory response	He et al. (2013)	
18	Human odontoblast-like cells contribute to inflammation by activating TLR2 in response to bacterial infection, leading to the release of pro-inflammatory cytokines such as IL-8 and IL-6	In vitro model of pulp inflammation: human odontoblast-like cells are cultured and stimulated with TLR2 agonists such as LTA or Pam2CSK4 to mimic inflammatory responses	Farges et al. (2011)	
19	TNF-α induces the expression of tissue-type plasminogen activator (tPA) and matrix metalloproteinases (MMPs) in human dental pulp cells and gingival fibroblasts, thereby contributing to pulp tissue degradation	TNF-α–HDP protease activation model: Human dental pulp-derived cells were cultured in vitro and stimulated with TNF-α to investigate its regulatory effects on PA and MMP activity	Ueda & Matsushima (2001)	
20	In pulpitis patients, HMGB1 levels are markedly elevated compared to healthy pulp tissue. HMGB1 interacts with the receptor for RAGE and enhances TLR2/4 signaling, while also promoting CpG DNA delivery and TLR9 activation, further driving cytokine production	HMGB1-induced inflammatory model in pulpitis-derived macrophages: Macrophages isolated from pulpitis patients were stimulated with HMGB1 in vitro, and molecular techniques were applied to investigate the regulatory role of HMGB1 on TLR2/TLR4 expression and proinflammatory cytokine release	Mahmoudi et al. (2017)	
21	Experimental studies have demonstrated that hydrogen peroxide (H2O2)-based bleaching agents generate ROS in dental pulp stem cells, resulting in cytotoxicity and nociceptive signaling	H2O2-induced cytotoxicity and pain transmission model in DPSCs: different concentrations of H2O2 bleaching gel are applied to enamel/dentin discs to stimulate DPSCs cultured on the dentin surface, aiming to investigate the mechanisms of H2O2-induced cytotoxicity and pain signaling	Chen et al. (2021)	
22	2-hydroxyethyl methacrylate (HEMA), a monomer used in restorative materials, can diffuse through dentinal tubules to the pulp, where it induces ROS production, eliciting inflammatory and cytotoxic responses	HEMA–TRPA1-mediated pain signaling model in dental pulp cells: human dental pulp stem cells (hDPSCs) are cultured and induced for osteogenic differentiation in vitro, then treated with various concentrations of HEMA to investigate how HEMA-induced ROS activates TRPA1 channels and promotes ATP release	Orimoto, Kitamura & Ono (2022)	

LPS may contribute to inflammation in infected pulp tissue by binding to TLRs and potentially activating necroptotic pathways involved in disease progression. Targeting this pathway—by using methods such as inhibiting LPS synthesis or blocking TLR signaling—could suppress excessive immune responses and tissue damage, offering a promising therapeutic strategy (Yan et al., 2022). Eritoran, a synthetic lipid A analogue, has demonstrated good safety and tolerability for topical use. By competitively binding to the TLR4/MD-2 complex, it exerts anti-inflammatory and analgesic effects in animal models through dose-dependent inhibition of pro-inflammatory cytokines. However, high systemic doses (e.g., 105 mg) did not enhance therapeutic outcomes, suggesting a limited effective dose range. To improve local efficacy and reduce systemic exposure, targeted delivery to the trigeminal ganglion or direct application within the root canal may be more effective strategies (Opal et al., 2013; Lin et al., 2015). Melatonin alleviates pulpal inflammation in a dose-dependent manner through a dual mechanism involving inhibition of the TLR4/NF-κB signaling pathway and enhancement of antioxidant enzyme activity, with low cytotoxicity (Li et al., 2015; Kermeoğlu et al., 2021; Kantrong, Jit-Armart & Arayatrakoollikit, 2020; Keskin, Şengül & Şirin, 2023). ML-193, a selective GPR55 antagonist, significantly reduces LPS-induced inflammatory mediator production in human dental pulp cells by directly blocking GPR55 and indirectly inhibiting the TLR4-MyD88-NF-κB pathway. In vitro studies show that ML-193 (0.5–10 µM) has minimal cytotoxicity, with optimal anti-inflammatory effects observed at 5–10 µM. Given its localized mechanism of action, intra-pulpal delivery of ML-193 may enhance therapeutic efficacy; however, further in vivo studies are needed to confirm its safety (Li & Shen, 2022; Lin et al., 2015). These results suggest that targeting necroptosis might represent a promising therapeutic approach to reduce the inflammation associated with pulpitis.

In the innate immune response, macrophages, neutrophils, and odontoblasts in the pulp serve as the first line of defense by rapidly recognizing pathogens and releasing pro-inflammatory mediators such as cytokines and chemokines. However, excessive release of DAMPs could trigger necroptosis, resulting in widespread immune cell death and the establishment of a self-perpetuating inflammatory cycle. Therefore, inhibiting the release of DAMPs may help suppress necroptosis-induced amplification of inflammation and promote the repair of pulp tissue. Inhibiting TNF-α has been shown to promote the proliferation, migration, and osteo/odontogenic differentiation of dental pulp stem cells (DPSCs), thereby facilitating dentin and bone regeneration (Zhan et al., 2022). HMGB1 inhibitors alleviate pulpitis by blocking its interaction with receptors such as RAGE and TLR4, leading to reduced pro-inflammatory cytokine production in a dose-dependent manner, as demonstrated by systemic intraperitoneal administration in animal studies. Although generally biocompatible, excessive HMGB1 inhibition may impair tissue repair, highlighting the need to balance anti-inflammatory efficacy with regenerative potential and evaluate long-term safety (Zhang et al., 2014; Diener et al., 2013; Qi et al., 2013). Thus, targeting necroptosis by inhibiting DAMPs like HMGB1 and pro-inflammatory cytokines such as TNF-α may reduce inflammatory mediators, disrupt the inflammatory cycle, and promote pulp tissue repair, offering a promising therapeutic strategy for pulpitis.

ROS, abundantly produced by immune cells during antibacterial responses, especially in bacterial infection-induced pulpitis, are primarily driven by mitochondrial dysfunction and represent a key contributor to disease progression. Stabilizing mitochondrial function to reduce ROS and inhibit necroptosis may help mitigate pulpitis. Core–shell chromium nanozymes (NanoCr) alleviate pulpal inflammation by scavenging ROS and restoring redox balance. While their efficacy is dose-dependent, potential particle aggregation at higher concentrations may influence their activity, warranting further investigation. Simultaneously, NanoCr exhibits relatively low cytotoxicity and good biocompatibility with dental pulp cells (Xie et al., 2023). Davallialactone exhibits antioxidant, anti-inflammatory, and reparative effects in vitro by suppressing ROS production in dental pulp cells, with effective activity at 10 µM and minimal cytotoxicity (Lee et al., 2011). Conceivably, inhibiting ROS production to block necroptosis could serve as a strategy to facilitate inflammation in pulpitis, simultaneously attenuating tissue damage and fostering more favorable disease progression (Table 3).

Table 3 Summary of therapeutic agents targeting necroptosis.

Drugs	Mechanism of action	Dose-response relationships	Administration routes	Biocompatibility issues	References	
Eritoran (TLR4 antagonist)	Eritoran, A synthetic lipid A analogue, inhibits the release of pro-inflammatory factors (such as TNF-α and IL-1β) in dental cc tissue by competitive binding to TLR4/MD-2	Eritoran has a dose-dependent anti-inflammatory and analgesic effect in animal models, but the clinical high dose (105 mg) has no additional benefit, suggesting that there is an effective dose range of its anti-inflammatory effect.	In order to improve the targeting, Eritoran was injected into the trigeminal ganglion of the affected teeth to block the pain-related TLR4 signal locally. Clinically, the drug can also be administered in the root canal or pulp cavity to achieve high concentration distribution of the lesion and reduce the systemic burden	Eritoran is well tolerated in human trials, and there is no significant difference in the incidence of adverse events between Eritoran and placebo in patients with sepsis. It has high biocompatibility and no obvious irritation reaction when applied topically, but its long-term effect still needs to be verified	Opal et al. (2013) and Lin et al. (2015)	
Melatonin	Melatonin can reduce the expression of pro-inflammatory factors and oxidative stress by scavenging ROS, inhibiting TLR4/NF-κB pathway and up-regulating antioxidant enzymes in the model of pulpitis, thus alleviating pulpitis	Animal experiments confirmed that melatonin had a dose-dependent anti-inflammatory effect, and intraperitoneal injection of 10 mg/kg could significantly reduce the inflammation of pulpitis. In vitro studies also showed that it enhanced the anti-inflammatory effect in the appropriate concentration range without damaging cell viability	In animal experiments, intraperitoneal injection is often used to systemically relieve pulpitis. In clinical practice, melatonin can be used as a pulp cap material or irrigation agent to act directly on dental pulp tissue and avoid the first-pass effect	As an endogenous hormone, melatonin has good biocompatibility. Low doses of melatonin can promote the proliferation of dental pulp cells, and high concentrations only slightly inhibit the proliferation. There are no serious adverse reactions in the clinical application of sleep aid, suggesting that it has a large safety margin in the local treatment of pulpitis	Li et al. (2015), Kermeoğlu et al. (2021), Kantrong, Jit-Armart & Arayatrakoollikit (2020) and Keskin, Şengül & Şirin (2023)	
ML-193 (GPR55 antagonist)	ML-193, a selective GPR55 antagonist, can block GPR55 and indirectly inhibit TLR4-MyD88-NF-κB pathway, reduce the production of IL-6, TNF-α, PGE2 and NO induced by LPS in human dental pulp cells, and exert a significant anti-inflammatory effect.	In vitro studies showed that ML-193 had no obvious cytotoxicity in the range of 0.5–10 μM, and could effectively inhibit LPS-induced inflammatory response in the range of 5–10 μM. Higher concentrations may enhance anti-inflammatory effects, but potential metabolic stress needs to be weighed against safety.	ML-193 is currently in the experimental stage, and there is no standard route of administration. Given the location of its target in the dental pulp, local delivery, such as endodontic or root canal administration, is considered feasible to reduce systemic side effects. The successful application of local injection in animal central nervous system models has been reported, suggesting its potential application in dental pulp local treatment in the future.	ML-193 shows no obvious in vitro cytotoxicity and can mitigate LPS-induced damage to dental pulp cells. Yet, its in-vivo safety, especially for systemic application, remains unevaluated. Currently, it is only applicable for local use in dental pulp tissue, and further animal studies are required to assess its biocompatibility.	Li & Shen (2022) and Li et al. (2015)	
HMGB1 Inhibitors (e.g., anti-HMGB1 antibody, glycyrrhizic acid, etc.)	HMGB1 inhibitor can inhibit the inflammatory cascade by neutralizing or blocking the binding of HMGB1 to RAGE, TLR4 and other receptors, reduce the production of pro-inflammatory factors such as IL-1β and CXCL1, down-regulate the expression of HMGB1 in T cells, inhibit the MAPK signaling pathway, and alleviate the inflammatory response of pulpium	Animal models have shown that the anti-inflammatory effect of HMGB1 inhibition is dose-dependent: high doses of anti-HMGB1 antibody or GA are more effective in inhibiting neutrophil infiltration, bone resorption and HMGB1/RAGE expression, and promoting the resolution of inflammation, while low doses are insufficient	In animal studies, intraperitoneal injection of anti-HMGB1 antibody or systemic administration of glycyrrhizic acid (GA) is often used to inhibit the high concentration of HMGB1 signal by reaching the inflammatory area of pulp through the blood. In the future, local infiltration injection can be explored, but it is necessary to ensure that the drug penetrates the pulp tissue	HMGB1 inhibitors such as neutralizing antibodies and glycyrrhizic acid are safe and well tolerated in animal experiments. However, because HMGB1 is involved in tissue repair, excessive inhibition or delay healing is observed. It is feasible to treat pulpitis, but it is still necessary to balance inflammation inhibition and regeneration and verify the long-term safety	Diener et al. (2013), Qi et al. (2013) and Xie et al. (2023)	
Core–shell chromium nanozymes (NanoCr nano-enzyme)	As an artificial nanoenzyme, core–shell chromium nanozymes can efficiently remove ROS, maintain cellular REDOX balance, and inhibit ROS-mediated inflammation. Moreover, NanoCr exhibited a broad spectrum inhibitory effect on a variety of pulpitis pathogens and reduced the pro-inflammatory effect of bacteria. NanoCr showed a synergistic anti-inflammatory effect in pulpitis through its dual mechanism of antioxidation and antibacterial	The efficacy of NanoCr was dose-dependent. At appropriate concentrations, chromium nanozymes exhibit optimal anti-inflammatory and antibacterial activities in an in vitro model. While their efficacy is dose-dependent, potential particle aggregation at higher concentrations may influence their activity, warranting further investigation. It has been proven to be effective in controlling inflammation and inhibiting major pathogenic bacteria within a specific dose range	NanoCr is mainly applied locally, and can be incorporated into pulp capping agent or root canal irrigating solution to directly act on the infected area. It is speculated that NanoCr gels can be implanted locally after dental pulp exposure to achieve sustained release and targeted therapy, while improving the efficacy and reducing systemic exposure	The biocompatibility of chromium nanozymes (NanoCr) is under evaluation. Preliminary cell experiments showed that NanoCr had no obvious toxicity and had protective effect on dental pulp cells at effective concentrations. As stable inorganic nanoparticles, attention should be paid to their long-term retention and the risk of metal ion release. At present, in vitro studies have shown good biocompatibility, and further animal experiments are needed to verify the long-term safety and metabolic characteristics in vivo	Xie et al. (2023)	
Davallialactone	Davallialactone has significant anti-inflammatory and reparative effects on dental pulp cells by reducing ROS production	The anti-inflammatory effects of davallialactone were dose-dependent, with higher concentrations more effectively suppressing LPS-induced inflammatory mediator expression and signaling pathway activation	Davallialactone was administered in vitro to cultured human dental pulp cells, either as a pretreatment or in combination with LPS, to evaluate its antioxidative and anti-inflammatory effects	Davallialactone exhibited good biocompatibility and showed no significant cytotoxicity toward human dental pulp cells.	Lee et al. (2011)	

This study aims to explore the therapeutic potential of targeting necroptosis to prevent the progression of pulpitis and its extension to the periapical region. Inhibiting the TLR4-RIPK3-mediated necroptotic pathway, which is activated by bacterial infection, can effectively reduce the release of DAMPs, thereby mitigating immune imbalance and dampening the amplification of inflammation. Concurrently, stabilizing mitochondrial function helps suppress excessive ROS production, alleviating oxidative stress and promoting tissue repair. Thus, targeting necroptosis may offer a promising strategy to disrupt the pathological interplay among bacterial infection, immune imbalance, and oxidative stress, thereby promoting the repair of pulpal tissue.

Study Limitations: Clinical Translational Challenges and Therapeutic Risks

Limitations of preclinical models in reflecting human pulpitis

Current investigations into the pathogenic mechanisms of necroptosis in pulpitis predominantly rely on murine models and in vitro cell-based systems, both of which have notable limitations that hinder mechanistic understanding and clinical translation. Anatomical and immunological disparities between rodent and human dental pulp, such as differences in dentinal tubule density and immune cell composition, limit the capacity of mouse models to replicate the chronic progression and microenvironmental complexity of human pulpitis.

For example, the hypoxic and acidic conditions typical of deep carious lesions in humans are poorly reproduced in rodent models. In vitro systems, such as LPS-stimulated dental pulp fibroblasts, often utilize single-pathogen stimuli and fail to capture the microbial diversity and host-pathogen interactions observed in vivo, including co-infections with organisms like Enterococcus faecalis (E. faecalis) and obligate anaerobes. Furthermore, the absence of single-cell sequencing data from necrotic human pulp tissue limits our understanding of species-specific expression patterns and post-translational modifications of key necroptotic mediators, such as RIPK3 isoforms or MLKL phosphorylation sites. Such gaps raise concerns that targeted therapies developed in preclinical models, such as RIPK1 inhibitors, may exhibit diminished efficacy or unintended off-target effects in clinical settings.

To address these challenges, future research should prioritize multi-omics analyses (e.g., transcriptomics, proteomics, phosphoproteomics) using human clinical samples, alongside the development of organoid-based infection models that better mimic the three-dimensional structure and immune landscape of dental pulp. These advanced systems can help validate findings from animal models and support the refinement of clinically translatable therapeutic strategies.

Excessive suppression of necroptosis: a risk to immune defense

Although necroptosis-targeting agents such as RIPK1 inhibitors have shown promise in reducing excessive inflammation, they may inadvertently impair the physiological immune functions of necroptosis. When intracellular pathogens such as viruses or E. faecalis evade apoptosis, necroptosis acts as a secondary defense by lysing infected cells, releasing PAMPs, and activating innate immune cells such as macrophages and dendritic cells. In oral infections, for instance, E. faecalis suppresses apoptosis to persist intracellularly, whereas RIPK3-MLKL-mediated necroptosis facilitates pathogen exposure and neutrophil-mediated clearance (Zou & Shankar, 2014). Excessive inhibition of necroptosis may suppress this protective response, increasing the risk of pathogen persistence, irreversible pulpitis, or bacteremia. Similar outcomes have been observed in viral infection models, where RIPK3-deficient hosts exhibit elevated viral loads and worsened neuroinflammation, as seen in herpes simplex virus type 1 (HSV-1) infection (Liu et al., 2023b).

Personalized design of necroptosis-targeted therapies

Given the pathogenic role of necroptosis in pulpitis and the limitations of current therapeutic approaches, future root canal treatment strategies should focus on improving localization, timing, and targeting precision. Localized delivery of RIPK1/RIPK3/MLKL pathway inhibitors may confine necroptosis suppression to the infection site, thereby reducing local inflammation without compromising systemic immune defenses. The timing of intervention is equally critical; during the early stages of infection, preserving moderate necroptotic activity may aid in pathogen clearance, whereas targeted inhibition during the inflammatory spread phase can limit tissue damage while sustaining immune protection. Dynamic monitoring of the inflammatory response, coupled with stage-specific modulation, may help optimize the balance between therapeutic efficacy and safety. Moreover, enhancing targeting specificity—such as through direct MLKL inhibition, development of more selective necroptosis modulators, or application of nano-delivery systems—could further refine treatment precision. These advancements are expected to improve root canal therapy outcomes by enhancing inflammation control, promoting pulp tissue preservation or regeneration, and achieving long-term therapeutic success.

Conclusion and Future Perspectives

Based on a comprehensive review of the existing literature, this review is the first to propose that the integration of bacterial infection, immune imbalance, and oxidative stress in necroptosis may contribute to the progression of pulpitis from reversible inflammation to irreversible necrosis. Bacterial infection in pulpitis may trigger necroptosis via the TLR4-RIPK3 pathway, with subsequent release of DAMPs disrupting immune balance and ROS from mitochondrial dysfunction exacerbating oxidative stress. These interacting mechanisms may collectively exacerbate pulpal inflammation and tissue damage, ultimately resulting in irreversible pulpitis.

Therefore, targeting necroptosis pathways may represent a promising therapeutic strategy for the management of pulpitis. We emphasized that future root canal therapies should place greater focus on individualized interventions, particularly in optimizing localized drug delivery, timing of intervention based on disease progression, and precise regulation guided by specific pathological mechanisms. These improvements aim to achieve a dynamic balance between inflammation control and pulp tissue preservation, thereby enhancing both the safety and effectiveness of treatment.

The authors wish to acknowledge Zhejiang University and Changsha Medical University for their support in scholarly discussion environment, access to literature/resources, and administrative coordination.

Additional Information and Declarations

Competing Interests

Author Contributions

Data Availability

The authors declare there are no competing interests.

Xuefei Wang conceived and designed the experiments, performed the experiments, analyzed the data, prepared figures and/or tables, authored or reviewed drafts of the article, and approved the final draft.

Yaying Wang conceived and designed the experiments, performed the experiments, analyzed the data, prepared figures and/or tables, authored or reviewed drafts of the article, and approved the final draft.

Zhenyu Pang conceived and designed the experiments, performed the experiments, analyzed the data, prepared figures and/or tables, authored or reviewed drafts of the article, and approved the final draft.

Peiyao Gong conceived and designed the experiments, performed the experiments, analyzed the data, prepared figures and/or tables, authored or reviewed drafts of the article, and approved the final draft.

Xuyi Ma conceived and designed the experiments, performed the experiments, analyzed the data, prepared figures and/or tables, and approved the final draft.

Yuhao Qiu conceived and designed the experiments, performed the experiments, prepared figures and/or tables, and approved the final draft.

Jianzhen Li conceived and designed the experiments, performed the experiments, prepared figures and/or tables, and approved the final draft.

Lianjie Peng conceived and designed the experiments, performed the experiments, analyzed the data, prepared figures and/or tables, authored or reviewed drafts of the article, and approved the final draft.

Zhichao Liu conceived and designed the experiments, performed the experiments, prepared figures and/or tables, authored or reviewed drafts of the article, and approved the final draft.

The following information was supplied regarding data availability:

Raw data was not generated in this literature review.

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
