# Peer review of "Role of necroptosis in pulpitis: integration of bacterial infection, immune imbalance, and oxidative stress"

_PeerJ, doi:10.7717/peerj.20209_

## Round 0.1 · original submission · Major Revisions

· Academic Editor

Major Revisions

**Language Note:** The review process has identified that the English language must be improved. PeerJ can provide language editing services - please contact us at [email protected] for pricing (be sure to provide your manuscript number and title). Alternatively, you should make your own arrangements to improve the language quality and provide details in your response letter. – PeerJ Staff

Reviewer 1 ·

Basic reporting

no comment

Experimental design

no comment

Validity of the findings

The article has the following shortcomings:
1、​Structural Monotony and Superficial Integration:
While the title emphasizes the "integration of bacterial infection, immune regulation, and oxidative stress," the content merely lists their individual pro-necroptotic effects without exploring their interrelationships or synergistic roles in driving pulpitis progression. The structure lacks depth. It is recommended to supplement analyses of how these three factors interact synergistically to amplify necroptosis.

2、​Detached Conclusions and Research Limitations:
The conclusions only broadly summarize that "necroptosis is a key mechanism in pulpitis" but fail to address the core question posed in the introduction regarding the integration of the three factors. The discussion neglects to contextualize how limitations in mechanistic understanding affect therapeutic translation.

3、​Insufficient Clinical Guidance:
Therapeutic strategies remain at a simplistic "mechanism → drug" extrapolation. There is a lack of personalized treatment design, and proposed therapies (e.g., melatonin) lack critical parameter discussions, including dose-response relationships, administration routes (local/systemic), and biocompatibility issues.

4、​Overemphasis on Pathological Effects:
The review disproportionately highlights necroptosis' detrimental role while overlooking its physiological functions in anti-infection immunity and tissue homeostasis. The proposal for "targeting necroptosis therapy" fails to discuss potential risks, such as impaired pathogen clearance due to excessive suppression of this protective mechanism.

Reviewer 2 ·

Basic reporting

The manuscript addresses a highly relevant and interdisciplinary topic, exploring the role of necroptosis in pulpitis and integrating it with bacterial infection, immune modulation, and oxidative stress. The language used is mostly clear and professional; however, several sections are overly verbose or repetitive, which may affect overall readability. The abstract is particularly long and could be significantly tightened to focus on key points and main contributions. The manuscript would benefit from a professional language edit to improve sentence structure and eliminate redundancy. Figures are appropriate in concept but could be more detailed and better labeled for clarity.

Experimental design

The review employs a systematic search strategy across four major databases, using relevant MeSH terms and Boolean logic. Inclusion/exclusion criteria are well defined. However, the manuscript does not include a summary table of included studies, which would enhance methodological transparency.

Validity of the findings

The conclusions presented are logical and consistent with the literature cited. The authors have successfully integrated molecular, immunological, and microbiological aspects of necroptosis in the context of pulpitis. However, certain claims—particularly those linking necroptosis directly to irreversible pulpitis and clinical progression—would benefit from more cautious language or supporting data. Include a paragraph discussing the limitations of current research on necroptosis in pulpitis, especially where data are derived from non-human models.

Additional comments

Add a summary table of pharmacological inhibitors or agents that target necroptosis (e.g., Necrostatin-1, melatonin) with mechanisms and references.
Include a brief discussion of potential risks or challenges in targeting necroptosis

·

Basic reporting

• The manuscript requires improvement in grammatical accuracy and overall language quality.
• What about the hypothesis proposed in the study?
• The aim of the study is not uniform in both the abstract and introduction; try to solve it.

Experimental design

• The manuscript requires improvement in grammatical accuracy and overall language quality.
• What about the hypothesis proposed in the study?
• The aim of the study is not uniform in both the abstract and introduction; try to solve it.

Validity of the findings

• The findings were not clear in the study.
• The type of the study was missed.
• What’s the novelty of the study?
• The abstract was not attractive and weak and needs to be repeated; it is nonstructural, so convert it into structural (background, aim of the study, methodology, results, conclusion). Also, the methodology of the result of the study was missing.

Additional comments

There are many comments about the manuscript:
1. Mechanisms of Necroptosis:
• The manuscript doesn’t clearly distinguish the cross-talk between apoptosis and necroptosis signaling.
• The term “microsomes" is introduced without context, which might confuse non-expert readers.
2. Pro-inflammatory Mechanisms of Necroptosis:
• The review mixes different pathways (TLR, TNFR1, and TRIF-RIPK3-MLKL) without clearly distinguishing between the extrinsic and intrinsic necroptosis triggers. Since the title focuses on the extrinsic pathway, a more focused discussion would strengthen the argument.
• The review lacks clear subheadings, making it difficult to follow complex processes like necrosome assembly, ROS generation, and mitochondrial dysfunction.
3. Necroptosis in Pulpitis:
• The review missed the real clinical trial data or case studies validating necroptosis-targeted therapies in human pulpitis treatment.
4. Conclusions:
• It was repetitive, and the conclusion of the abstract was poor. Replace it.

---

## Round 0.2 · accepted · Accept

· Academic Editor

Accept

I have confirmed that the authors have addressed all of the reviewers' comments, and therefore, I consider this manuscript ready for publication.